# Tumour-specific PI3K inhibition via nanoparticle-targeted delivery in head and neck squamous cell carcinoma

Aviram Mizrachi[1,2,*,†], Yosi Shamay[3,*], Janki Shah[3], Samuel Brook[4], Joanne Soong[4], Vinagolu K. Rajasekhar[2], John L. Humm[5], John H. Healey[2], Simon N. Powell[6], José Baselga[4,7], Daniel A. Heller[3,8], Adriana Haimovitz-Friedman[6] & Maurizio Scaltriti[4,9]

Alterations in *PIK3CA*, the gene encoding the p110α subunit of phosphatidylinositol 3-kinase (PI3Kα), are frequent in head and neck squamous cell carcinomas. Inhibitors of PI3Kα show promising activity in various cancer types, but their use is curtailed by dose-limiting side effects such as hyperglycaemia. In the present study, we explore the efficacy, specificity and safety of the targeted delivery of BYL719, a PI3Kα inhibitor currently in clinical development in solid tumours. By encapsulating BYL719 into P-selectin-targeted nanoparticles, we achieve specific accumulation of BYL719 in the tumour milieu. This results in tumour growth inhibition and radiosensitization despite the use of a sevenfold lower dose of BYL719 compared with oral administration. Furthermore, the nanoparticles abrogate acute and chronic metabolic side effects normally observed after BYL719 treatment. These findings offer a novel strategy that could potentially enhance the efficacy of PI3Kα inhibitors while mitigating dose-limiting toxicity in patients with head and neck squamous cell carcinomas.

[1] Head and Neck Service, Memorial Sloan Kettering Cancer Center, New York, New York 10065, USA. [2] Department of Surgery, Memorial Sloan Kettering Cancer Center, New York, New York 10065, USA. [3] Molecular Pharmacology Program, Memorial Sloan Kettering Cancer Center, New York, New York 10065, USA. [4] Human Oncology & Pathogenesis Program, Memorial Sloan Kettering Cancer Center, New York, New York 10065, USA. [5] Department of Medical Physics, Memorial Sloan Kettering Cancer Center, New York, New York 10065, USA. [6] Department of Radiation Oncology, Memorial Sloan Kettering Cancer Center, New York, New York 10065, USA. [7] Department of Medicine, Memorial Sloan Kettering Cancer Center, New York, New York 10065, USA. [8] Department of Pharmacology, Weill Cornell Medical College, New York, New York 10065, USA. [9] Department of Pathology, Memorial Sloan Kettering Cancer Center, New York, New York 10065, USA. * These authors contributed equally to this work. † Present address: Department of Otorhinolaryngology Head and Neck Surgery, Rabin Medical Center, 39 Jabotinski St., Petah Tikva 49100, Israel. Correspondence and requests for materials should be addressed to D.A.H. (email: hellerd@mskcc.org) or to A.H.-F. (email: a-haimovitzfriedman@ski.mskcc.org) or to M.S. (email: scaltrim@mskcc.org).

In head and neck squamous cell carcinoma (HNSCC), activating mutations or copy number gain of *PIK3CA* are the most frequent therapeutically relevant aberrations[1–4]. In the clinical setting, however, the mainstay treatment for HNSCC is surgery and/or radiation therapy (RT), administered concomitantly with chemotherapy for radio-sensitization[5,6]. Recent pre-clinical data from our group and others suggest that inhibition of the PI3K/AKT/mTOR pathway acts as a potent radio-sensitizer in *PIK3CA*-mutated HNSCC[7–10]. As PI3Kα inhibitors have shown promising activity in *PIK3CA*-mutant tumours[11,12], these agents could potentially substitute chemotherapy in a significant proportion of this patient population. Moreover, PI3K inhibition has been shown to enhance the expression of *TP53* in HNSCC and inhibit tumour growth either alone or in combination with RT[13]. However, a relatively narrow therapeutic window that inevitably leads to emergence of toxicity limits the clinical use of PI3K and AKT inhibitors. Side effects include skin toxicity, diarrhoea, fatigue and, in virtually all patients, hyperglycaemia[14]. Therefore, dose limitations for systemic administration of PI3K inhibitors often result in suboptimal concentrations that insufficiently engage the target failing to produce lasting anti-tumour efficacy[15,16].

Targeted drug delivery systems aim to improve therapeutic efficacy by increasing the tumour concentration of anti-cancer drugs while reducing drug exposure in disease-free organs[17–19]. We recently developed a novel drug delivery system by exploiting the nanomolar affinity of fucoidan polysaccharide[20] to the cell adhesion molecule P-selectin[21,22]. Upon endothelial activation with endogenous cytokines, or exogenous stimuli such as ionizing radiation, P-selectin translocates to the cell membrane and into the lumen of blood vessels. Importantly, high levels of P-selectin have been found in the vasculature of several human cancers[20,23–25]. In the present study, we aimed to test the performance of fucoidan-based nanoparticles in delivering the PI3Kα inhibitor BYL719 (Novartis Pharmaceuticals)[26] in the tumour milieu of HNSCC. We demonstrate that tumour-specific P-selectin-dependent accumulation of BYL719 can suppress tumour growth without the emergence of on-target adverse effects due to systemic drug administration.

## Results

**Characterization of HNSCC models**. Upon analysing the tumour microvasculature of HNSCC models established in our laboratory, we found that both cell line-based tumours and patient-derived xenografts (PDXs) showed strong staining for P-selectin (Supplementary Fig. 1a,b). Using MSK-Integrated Mutation Profiling of Actionable Cancer Targets (IMPACT[TM]), a deep-coverage-targeted sequencing analysis of 410 key cancer-associated genes[27]; we sequenced these tumours and confirmed the presence of common genetic alterations typical of HNSCC, including *PIK3CA* activating mutations (Supplementary Table 1). To study the efficacy of P-selectin-targeted PI3Kα inhibition *in vivo*, we selected one HNSCC cell line model (Cal-33) and one PDX model (H22) obtained from a patient with previously untreated advanced laryngeal cancer. Both tumours harbour a *PIK3CA* hotspot activating mutation (missense H1047R) and express high levels of P-selectin. No *PIK3CA* wild-type models were used, as they are known to be generally refractory to PI3Kα inhibition[28].

**Nanoparticle preparation and targeting**. Fucoidan-based nanoparticles containing BYL719 (FiBYL719) were prepared by co-encapsulating both the drug and a near-infrared dye (IR820) to facilitate imaging. As a negative control for targeting studies, we prepared drug-loaded dextran sulfate-based nanoparticles (DexBYL719) that lacked fucoidan (Supplementary Fig. 2). We previously found that dextran sulfate-based particles did not bind to P-selectin but could passively target tumours, likely via the enhanced permeability and retention effect[20]. These control nanoparticles exhibited comparable physical properties to those of FiBYL719 and were assembled using the same procedures (Supplementary Fig. 3a–d). We then measured the drug release profiles of BYL719 from FiBYL719 nanoparticles at pH 5.5 and 7.4 (Supplementary Fig. 3e). Drug release accelerated substantially at low pH. Finally, we assessed the *in vitro* binding affinity of FiBYL719 and control DexBYL719 nanoparticles to bovine aortic endothelial cells stimulated to express P-selectin with either tumour necrosis factor α (TNFα) or RT. As expected, only FiBYL719 nanoparticles penetrated into the endothelial cells upon stimulation (Supplementary Fig. 3f).

We administered the nanoparticles in nude mice bearing subcutaneous H22 PDX tumours. After 24 h, we found a significantly higher tumour localization of FiBYL719 nanoparticles compared with DexBYL719 nanoparticles (Figs 1a,b). When the animals were pre-treated with a P-selectin blocking antibody, the localization of FiBYL719 nanoparticles in the tumour was abrogated (Fig. 1a,b).

Upon irradiation of Cal-33 xenograft-bearing mice with a dose of 4 Gy, we found an enhancement of P-selectin expression in the tumour vasculature (Fig. 1c,d). Administration of FiBYL719 nanoparticles into the irradiated mice resulted in increased drug accumulation (Fig. 1e,f) and specific localization of the nanoparticles in the tumour microenvironment (Fig. 1g) as evinced by fluorescence microscopy.

**Evaluation of FiBYL719 anti-tumour efficacy**. We then determined whether tumour accumulation of FiBYL719 nanoparticles translated in PI3K/AKT/mTOR pathway inhibition in HNSCC tumours. We treated Cal-33 tumour-bearing mice with a single administration of BYL719, either in the form of the free drug (50 mg kg$^{-1}$), the standard dose given PO in mice[29], or encapsulated into fucoidan nanoparticles (25 mg kg$^{-1}$), the maximal dose we were able to encapsulate and give intravenously. We used S6 ribosomal protein (S6) phosphorylation as a readout of the pharmacodynamics of the inhibitor, as this marker integrates the effects of BYL719 on both PI3K/AKT and mTORC1[29]. Treatment with free BYL719 elicited a strong albeit transient inhibition of the pathway, which was partially restored after 6 h and fully restored by 24 h, compatible with the relatively short half life of BYL719 in plasma[28]. In contrast, FiBYL719 resulted in complete and durable suppression of S6 phosphorylation over 24 h (Fig. 2a). Western blot analysis of the same xenografts confirmed the lasting inhibition of S6 phosphorylation and showed concomitant activation of pERK (Fig. 2b), a well-known feedback mechanism triggered by suppression of the PI3K/AKT pathway[30–32]. These findings were further confirmed in a 3-D reconstruction of an immunofluorescence analysis of two representative Cal-33 tumours collected at 24 h post treatment. In the tissue section of the tumour treated with FiBYL719, we observed diminished staining for pS6 and increased apoptosis, denoted by caspase 3 cleavage, compared with the tumour treated with oral BYL719 (Fig. 2c,d).

*In vivo* efficacy studies were conducted in both Cal-33 and H22 PDX models. Mice were randomized into four treatment arms: vehicle control, free BYL719 administered 7 mg kg$^{-1}$ per day (total 50 mg kg$^{-1}$ per week), free BYL719 administered 50 mg kg$^{-1}$ per day (total 350 mg kg$^{-1}$ per week) and nanoparticle-encapsulated FiBYL719 administered 25 mg kg$^{-1}$ twice a week (total 50 mg kg$^{-1}$ per week). Significant tumour

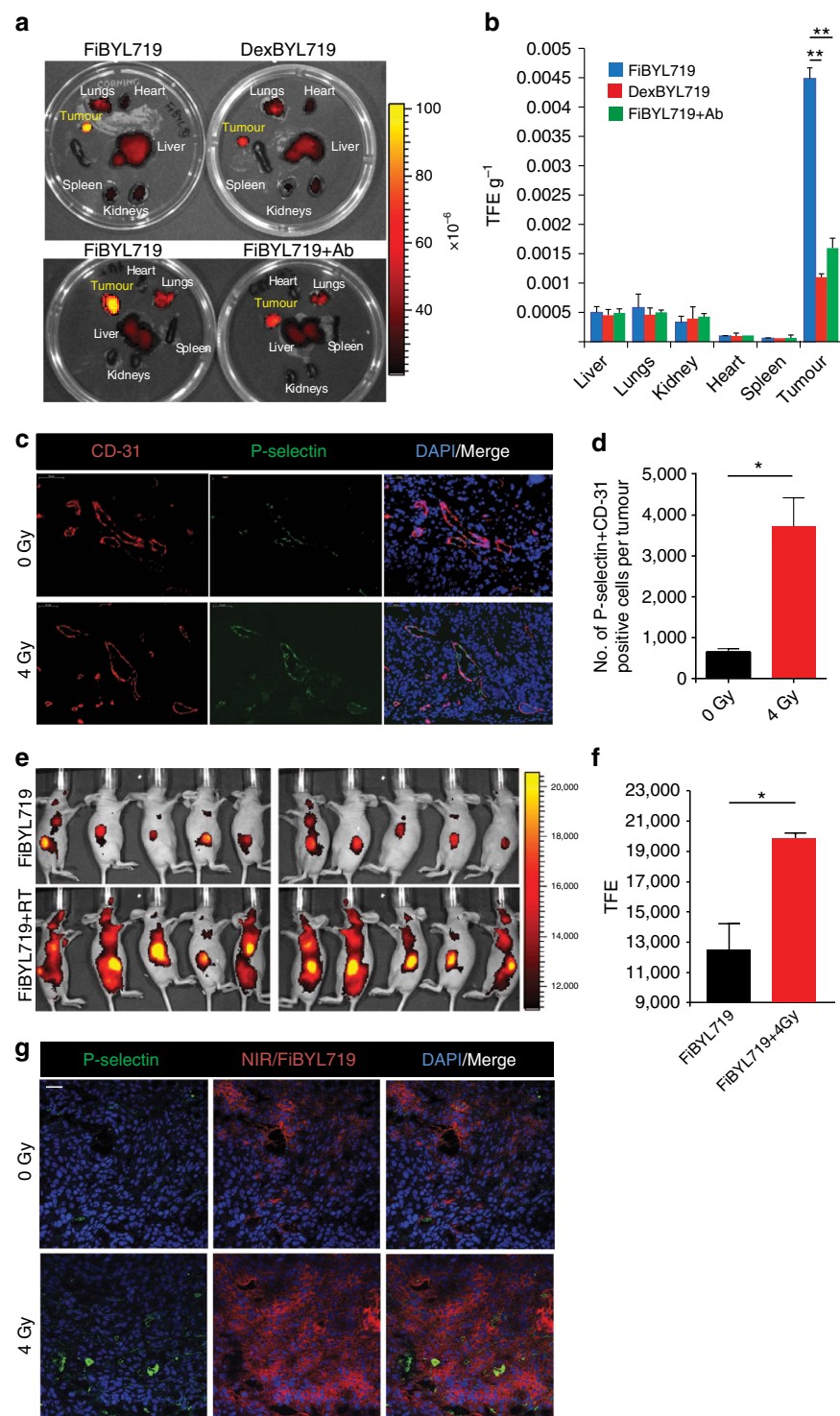

**Figure 1 | *In-vivo* targeting of BYL719-loaded nanoparticles prepared with either fucoidan (Fi) or dextran sulfate (Dex).** (**a**) Representative *ex vivo* fluorescence images of mice organs 24 h after i.v. administration of FiBYL719 or DexBYL719 nanoparticles, and pre-treated with anti-P-selectin antibody (Ab). (**b**) Nanoparticle biodistribution in organs and tumour, calculated from *ex vivo* fluorescence images shown in **a** as total fluorescence efficiency divided by organ weight (*n* = 3). (**c**) Representative immunofluorescence staining for CD-31 (red), P-selectin (green) and DAPI (blue) in Cal-33 xenografts before and after a single dose of ionizing radiation (4 Gy). Scale bars, 50 μm. (**d**) Quantification of double-staining positive endothelial cells per tumour shown in **c** (*n* = 3). (**e**) *In vivo* fluorescence imaging of Cal-33 xenograft-bearing mice 24 h after treatment with FiBYL719 or 4 Gy RT followed by FiBYL719. (**f**) Quantification of total fluorescence efficiency of tumours shown **e** (*n* = 10). (**g**) Representative immunofluorescence stains of tumour sections for P-selectin (green), NIR (red) and DAPI (blue) from H22 xenografts 24 h after treatments. Scale bars, 50 μm. In **b,d,f** error bars indicate mean ± s.e.m. *P < 0.05, **P < 0.01; by Mann–Whitney *U*-test in **b** or one-way ANOVA with *post hoc* Tukey test in **d,f**.

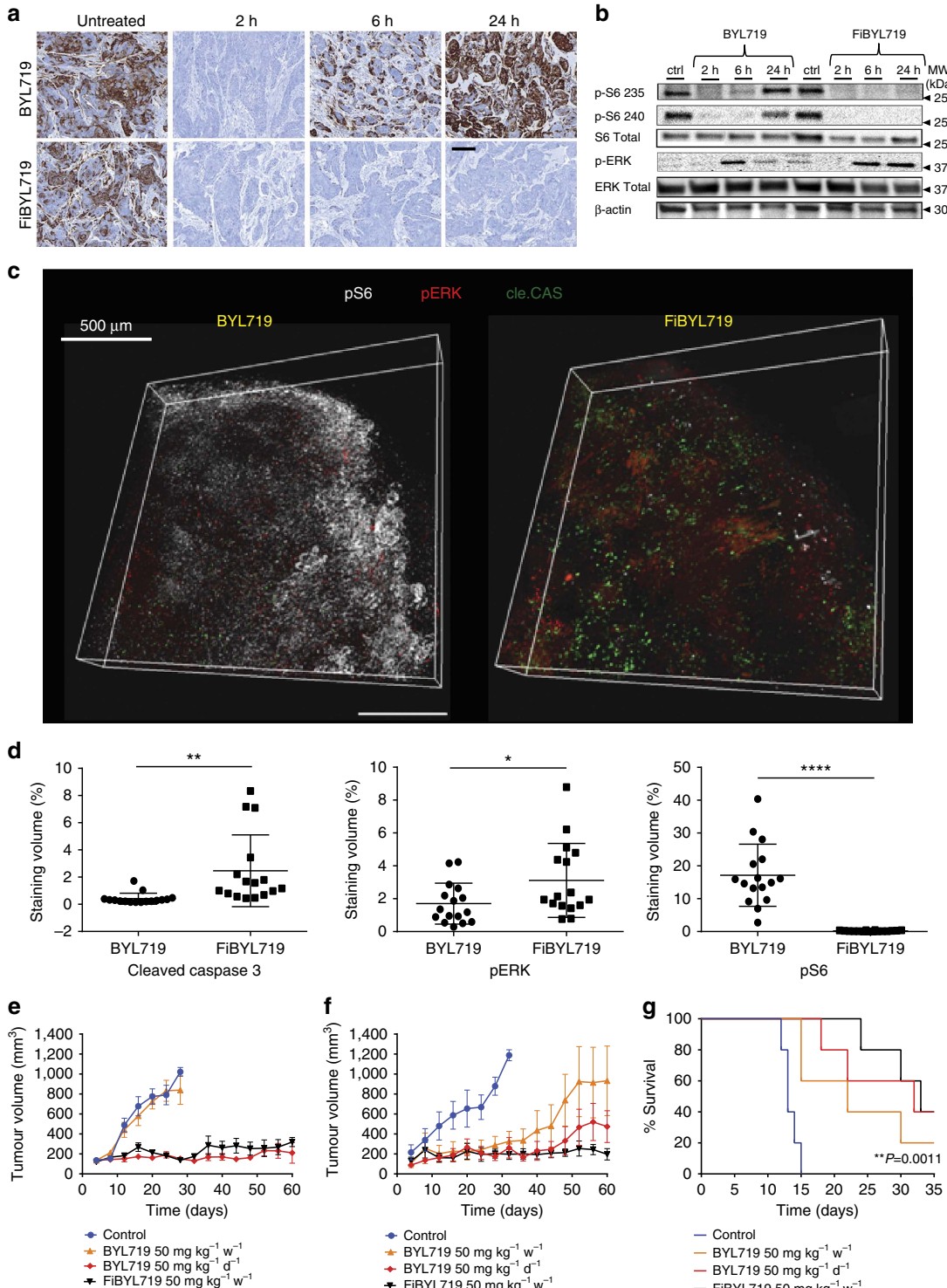

**Figure 2 | Anti-tumour efficacy of free BYL719 and nanoparticle-encapsulated FiBYL719 in pre-clinical HNSCC models. (a)** Representative images of immunohistochemistry staining for pS6 at different time points following treatment with 25 mg kg$^{-1}$ BYL719 or 25 mg kg$^{-1}$ FiBYL719 in Cal-33 xenografts ($n=3$). Scale bar, 50 μm. **(b)** Western blot of pS6 and pERK in Cal-33 xenograft tissues following treatment with 25 mg kg$^{-1}$ BYL719 or 25 mg kg$^{-1}$ FiBYL719, $n=3$. **(c)** 3-D reconstruction of a stained Cal-33 xenograft section 24 h after treatment with either 50 mg kg$^{-1}$ BYL719 or 25 mg kg$^{-1}$ FiBYL719. pS6 (white), pERK (red) and cleaved caspase 3 (green). Scale bar, 500 μm. **(d)** Box plots comparing the volume of positive staining (% of total tissue volume) of tissues shown in **c**, ($n=2$). **(e)** Tumour growth curves of Cal-33 xenografts treated with oral administration of either 50 or 7 mg kg$^{-1}$ BYL719 daily, or i.v. injection of 25 mg kg$^{-1}$ FiBYL719 bi-weekly ($n=10$). **(f)** Tumour growth curves of H22 patient-derived xenografts treated with oral administration of either 50 or 7 mg kg$^{-1}$ BYL719 daily, or bi-weekly i.v. injections of 25 mg kg$^{-1}$ FiBYL719 ($n=10$). **(g)** Survival curve of mice engrafted with orthotopic tongue cal-33 xenografts treated with oral administration of either 50 or 7 mg kg$^{-1}$ BYL719 daily or i.v. injections of 25 mg kg$^{-1}$ FiBYL719 bi-weekly ($n=5$). In **d**–**f**, error bars indicate mean ± s.e.m. *$P<0.05$, **$P<0.01$, ****$P<0.0001$; by one-way ANOVA with *post hoc* Tukey test. In **g**, $P$-value was calculated by using the log-rank test.

inhibition was observed in both Cal-33 and H22 models upon administration of FiBYL719 nanoparticles. The anti-tumour effects of a weekly dose of nanoparticles were comparable to those of a sevenfold higher dose of the un-encapsulated free drug. The equivalent dose of free BYL719 administered at $7 \, \mathrm{mg \, kg^{-1}}$ per day ($50 \, \mathrm{mg \, kg^{-1}}$ per week) elicited no appreciable anti-tumour activity in Cal-33 tumours (Fig. 2e), whereas in H22-bearing mice it resulted in transient delay of tumour growth followed by acquired insensitivity to the treatment (Fig. 2f).

We next tested the ability of FiBYL719 nanoparticles to accumulate and exert their anti-tumour activity in an orthotopic HNSCC mouse model. Cal-33 cells were engrafted into the tongue of nude mice and, at 7 days post injection, obvious tongue tumours developed (Supplementary Fig. 4a,b), and metastases to cervical lymph nodes appeared within 1–2 weeks (Supplementary Fig. 4c,d). The orthotopic tongue tumours also exhibited positive P-selectin expression in the associated vasculature (Supplementary Fig. 4e). Consistent with the data from the subcutaneous models, we observed that the mean survival of mice treated with $25 \, \mathrm{mg \, kg^{-1}}$ FiBYL719 twice a week ($50 \, \mathrm{mg \, kg^{-1}}$ per week) or daily free $50 \, \mathrm{mg \, kg^{-1}}$ BYL719 ($350 \, \mathrm{mg \, kg^{-1}}$ per week) were similar (33 days). The mean survival of mice treated with free $7 \, \mathrm{mg \, kg^{-1}}$ BYL719 daily ($50 \, \mathrm{mg \, kg^{-1}}$ per week) was relatively short (22 days) as well as that of the vehicle control group (13 days) (Fig. 2g).

**Combination of FiBYL719 and radiation**. We investigated the effects of RT on P-selectin-targeted PI3Kα inhibition. We reasoned that increased efficacy may result from the combined effects of RT to increase nanoparticle localization to the tumour (Fig. 1e,f) and of PI3Kα inhibition to sensitize HNSCC to RT[7–10]. First, we measured the effects of applying a single dose of 4 Gy RT to H22 tumour-bearing mice in combination with $25 \, \mathrm{mg \, kg^{-1}}$ FiBYL719 nanoparticles or $50 \, \mathrm{mg \, kg^{-1}}$ free BYL719. Approximately 24 h after treatment, we found that tumour γH2AX nuclear foci formation, an indicator of DNA damage, was significantly augmented upon treatment with the nanoparticles as compared with the free drug or RT alone (Fig. 3a,b). Apoptosis in the tumour tissue was also substantially increased by the nanoparticles, as measured by cleaved poly ADP ribose polymerase (PARP) expression (Fig. 3c) and confirmed by TdT-mediated dUTP nick end labelling (TUNEL) staining (Fig. 3d).

We then investigated whether the nanoparticle/RT combination could produce long-term inhibition of tumour growth in the H22 PDX model. A clinically relevant dose of fractionated RT (4 Gy × 5) was administered alone or in combination with free BYL719 dosed at $7 \, \mathrm{mg \, kg^{-1}}$ per day, free BYL719 at $50 \, \mathrm{mg \, kg^{-1}}$ per day or FiBYL719 at $25 \, \mathrm{mg \, kg^{-1}}$ administered twice per week. As previously reported[7], treatment without nanoparticle encapsulation was sufficient to delay tumour growth to some extent. However, we found that only 5 days of treatment with FiBYL719 (two single administrations of $25 \, \mathrm{mg \, kg^{-1}}$) were sufficient to achieve durable stabilization of all tumours, as compared with free drug or RT alone (Fig. 3e).

**Amelioration of side effects**. Upon systemic treatment with PI3K/AKT inhibitors, hyperglycaemia is induced by phosphorylation of insulin receptor (IR) leading to loss of insulin signalling in peripheral tissue and pancreatic β cells[33,34]. To assess whether P-selectin-mediated-targeted delivery of BYL719 could prevent systemic drug exposure, we measured serum glucose and insulin levels in healthy mice treated with either BYL719 or FiBYL719. A single dose of free BYL719 (either 25 or $50 \, \mathrm{mg \, kg^{-1}}$) resulted in a spike in serum glucose and insulin levels 1–8 h after

treatment. Upon administration of nanoparticle-encapsulated $25 \, \mathrm{mg \, kg^{-1}}$ FiBYL719, only a slight increase of glucose levels was observed, and no effect on insulin levels was detectable within 24 h (Fig. 4a,b).

To benchmark PI3K inhibition in insulin target organs, we measured the phosphorylation status of IR and the glucose metabolism regulatory enzyme glycogen synthase kinase-3 β (GSK3β)[35] upon treatment with either $50 \, \mathrm{mg \, kg^{-1}}$ free BYL719 or $25 \, \mathrm{mg \, kg^{-1}}$ FiBYL719. Administration of free BYL719 resulted in rapid phosphorylation of IR and inhibition of phosphorylation of GSK3β, whereas neither IR nor GSK3β phosphorylation status were affected by FiBYL719 (Fig. 4c).

We next sought to evaluate whether continuous treatment with FiBYL719 nanoparticles could also obviate the chronic effects of prolonged PI3K inhibition on glucose metabolism[36]. Mice were treated for 60 consecutive days with either BYL719 ($50 \, \mathrm{mg \, kg^{-1}}$ per day) or FiBYL719 nanoparticles ($50 \, \mathrm{mg \, kg^{-1}}$ per week). Treatments were then halted for 72 h before blood and pancreas samples were collected for analysis. We found significantly elevated serum glucose and insulin levels in the mice treated with free BYL719 but not in the FiBYL719-treated group (Fig. 4d,e). Moreover, a lower number of insulin-producing β cells per islet and a higher number of glucagon-producing α cells per islet were detected in the free BYL719-treated versus FiBYL719-treated animals (Fig. 4f–h).

Finally, we assessed systemic toxicity by analysing the histology of heart, kidney, spleen, lungs and liver of animals treated with FiBYL719 (Fig. 5a). No morphologic changes suggestive of tissue damage were observed after 14 days of drug administration. Consistently, no inhibition of the PI3K/mTOR pathway was observed in the same tissues (Fig. 5b). Moreover, we observed a normal complete blood count (Supplementary Table 3) and no evidence of body weight loss of the treated animal (Fig. 5c), suggesting that there were no nanotherapeutic-related toxicities.

## Discussion

Anti-tumour kinase inhibitors have become a standard of care due to their specificity and selectivity to unique genomic aberrations present in certain malignancies. However, most of these compounds only lead to transient inhibition of their targets, necessitating daily or weekly administration in order to achieve clinically effective intratumoural drug concentrations. Depending on systemic treatment duration, the amount of drug needed to efficaciously inhibit the target often yields off-target and on-target effects on healthy tissues and causes intolerable adverse effects due to systemic exposure. A narrow 'therapeutic window' represents the main limitation for the anti-tumour activity of virtually any kinase inhibitor administered systemically.

Activating mutations or amplification of *PIK3CA*, the gene encoding the class IA PI3K catalytic subunit p110α, is the most common genomic alteration in HNSCC, present in up to 40% of human papilloma virus-positive cases. Specific PI3Kα inhibitors are under current investigation in both pre-clinical and clinical settings of HNSCC[7,31].

In the present study, we exploited the facts that P-selectin is expressed in HNSCC tumour milieu and can be upregulated by radiation to test the efficacy of P-selectin-mediated delivery of a specific PI3Kα inhibitor, BYL719, using fucoidan-based nanoparticles in models of HNSCC. The goal of this work was to investigate whether the specific accumulation of BYL719 in the tumour microenvironment is sufficient to exert significant anti-tumour effects while sparing healthy tissues from systemic exposure and related toxicities. We found that FiBYL719

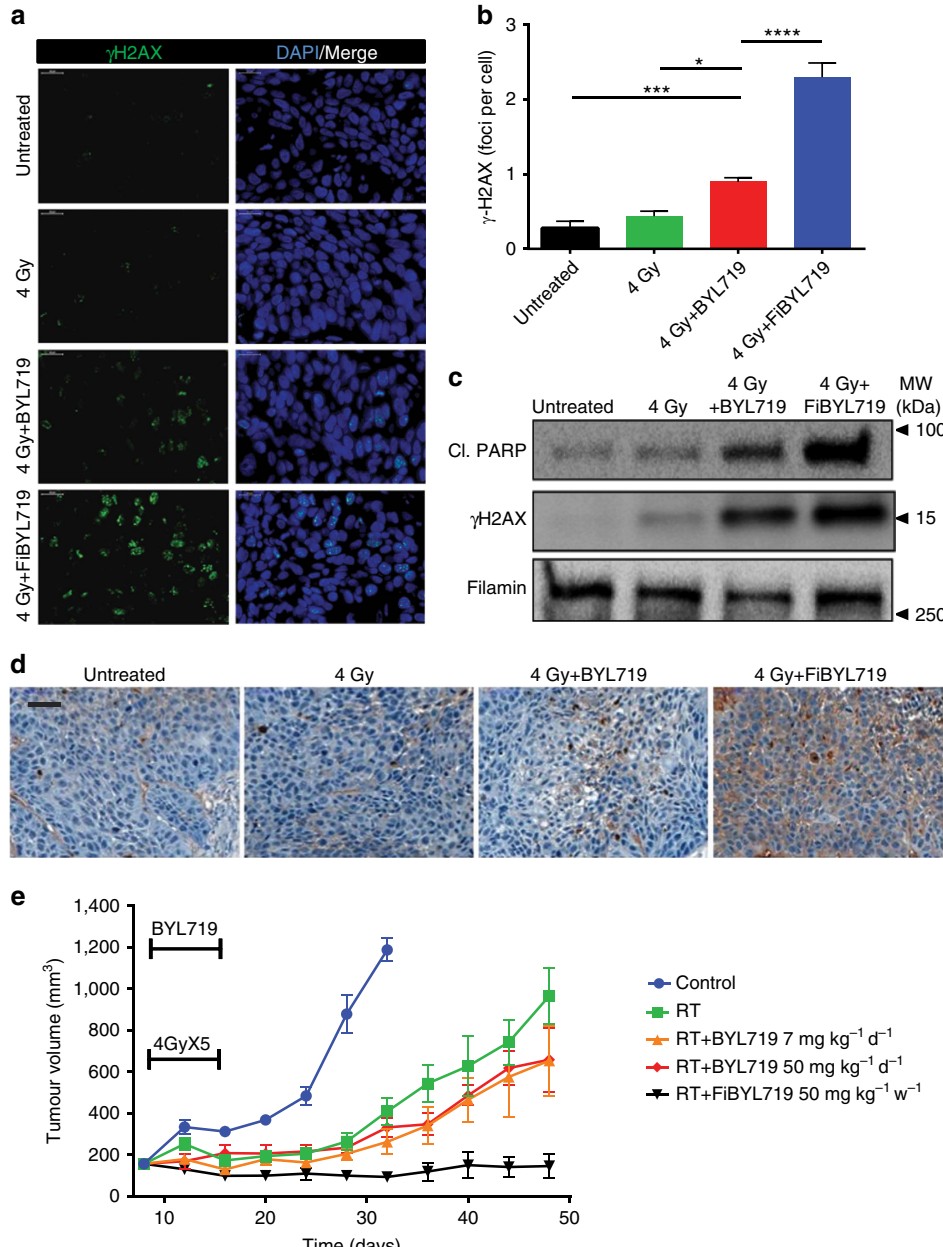

**Figure 3 | Radiosensitization effects of pre-clinical HNSCC models by free and nanoparticle-encapsulated BYL719. (a)** Representative images of immunofluorescence staining for nuclear γH2AX foci (green) and DAPI (blue) in H22 patient-derived xenografts 24 h post treatment with RT (4 Gy) or RT followed by 50 mg kg$^{-1}$ BYL719 or 25 mg kg$^{-1}$ FiBYL719. Scale bar, 20 μm. **(b)** Quantification of γH2AX staining (foci per cell) presented in **a** ($n = 3$). **(c)** Western blot of γH2AX and cleaved PARP in H22 patient-derived xenografts 24 h post treatment with RT (4 Gy), RT and 50 mg kg$^{-1}$ BYL719, or 25 mg kg$^{-1}$ FiBYL719 ($n = 3$). **(d)** Representative images of immunohistochemical staining for TUNEL in H22 patient-derived xenografts 24 h post treatment with 4 Gy RT, 4 Gy RT plus 50 mg kg$^{-1}$ BYL719, or 4 Gy RT plus 25 mg kg$^{-1}$ FiBYL719. Scale bar, 50 μm. **(e)** Tumour growth curves of H22 patient-derived xenografts treated for 5 days with daily oral administration of either 50 or 7 mg kg$^{-1}$ BYL719 daily, or with i.v. injections of 25 mg kg$^{-1}$ FiBYL719 administered bi-weekly, combined with fractionated RT of 4 Gy × 5 doses on days 1–5 ($n = 10$). In **b,e**, error bars indicate mean ± s.e.m. *$P < 0.05$ ***$P < 0.001$, ****$P < 0.0001$; by one-way ANOVA with *post hoc* Tukey test.

administration leads to prolonged and tumour-specific inhibition of the PI3K/AKT/mTOR pathway, which resulted in durable control of tumour growth. These effects were enhanced by concomitant RT treatment, presumably due to both DNA damage induced by PI3K inhibition, increased P-selectin-mediated FiBYL719 tumour accumulation and prolonged PI3K pathway inhibition. This reverberating effect is particularly relevant in HNSCC, where RT therapy is the standard of care.

Upon systemic treatment with PI3K/AKT inhibitors, hyperglycaemia is invariably induced by loss of insulin signalling in peripheral tissue and pancreatic β cells[33,34]. Thus, an acute increase of glucose and insulin in the blood stream can be used as a readout of systemic drug exposure and engagement of PI3K in healthy tissues. Accordingly, a rapid spike in both glycaemia and insulinaemia was observed in mice following oral administration of BYL719, whereas these effects were largely attenuated by targeted delivery of BYL719 using fucoidan nanoparticles.

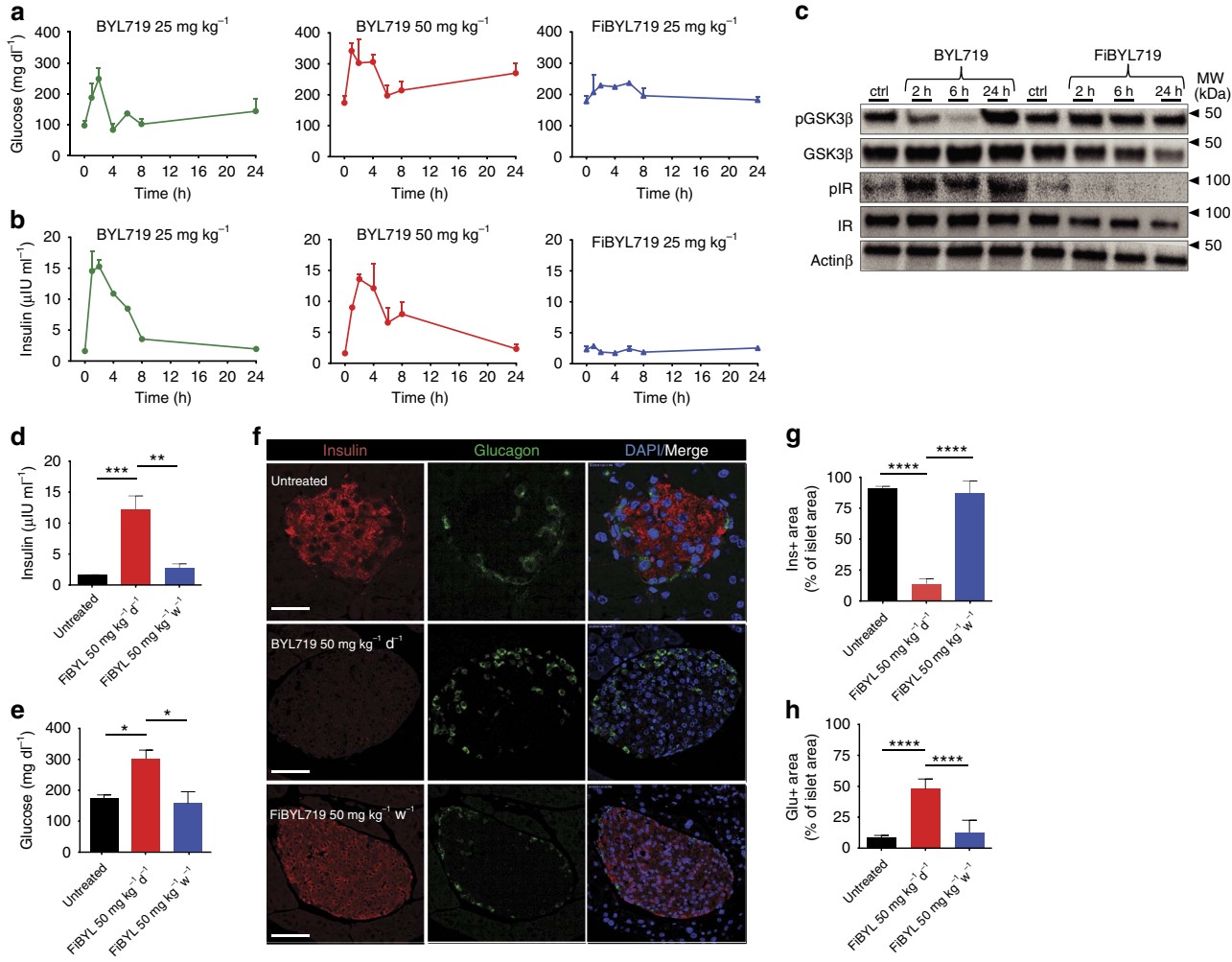

**Figure 4 | Amelioration of systemic metabolic effects of PI3K inhibition by P-selectin-targeted delivery of BYL719.** (**a,b**) Serum glucose and insulin levels of mice treated with 25 and 50 mg kg$^{-1}$ BYL719 or 25 mg kg$^{-1}$ FiBYL719 ($n = 6$). (**c**) Western blot of pGSK and pIR in mice livers following treatment with 50 mg kg$^{-1}$ BYL719 or 25 mg kg$^{-1}$ FiBYL719 ($n = 6$). (**d,e**) Serum insulin and glucose levels of mice following 60 days of treatment with 50 mg kg$^{-1}$ BYL719 daily or 25 mg kg$^{-1}$ FiBYL719 bi-weekly ($n = 6$). (**f**) Representative images of immunofluorescence staining for insulin (red), glucagon (green) and DAPI (blue) of mice pancreatic islets following 60 days of treatment with 50 mg kg$^{-1}$ BYL719 administered daily, or 25 mg kg$^{-1}$ FiBYL719 administered bi-weekly ($n = 3$). Scale bar, 50 μm. (**g,h**) Bar graph showing the percentage of insulin or glucagon positive staining areas of total tissue area from pancreatic islets ($n = 3$). In **d,e,g,h**, error bars indicate mean ± s.e.m. *$P < 0.05$, **$P < 0.01$, ***$P < 0.001$, ****$P < 0.0001$; by one-way ANOVA with *post hoc* Tukey test.

In addition to these transient modulations of glucose metabolism, morphological changes in the pancreas have been described in mice subjected to chronic hyperglycaemia and hyperinsulinemia that mimics non-insulin-dependent diabetes mellitus, characterized by insulin resistance and glucose intolerance[37]. Our findings suggest that FiBYL719 treatment can produce durable tumour-specific inhibition of the PI3K pathway without the emergence of chronic hyperglycaemia and hyperinsulinaemia that results in exhaustion of the insulin-producing β cells and compensatory augmentation of glucagon-producing α cells.

It remains to be elucidated whether concomitant inhibition of ERK by either systemic or nanoparticle-based treatment with MEK inhibitors would further enhance the anti-tumour activity of BYL719. Previous reports testing the efficacy of PI3K/mTOR blockade in combination with mitogen-activated protein kinase (MEK) inhibition in HNSCC[32] and other tumour models[30,38] suggest that this may be the case. We have recently showed that a selective MEK inhibitor encapsulated in P-selectin-targeted nanoparticles exhibited improved efficacy over the free drug with lower dosing and prolonged target engagement in the tumour compared with the skin, a major site of on-target toxicity[20]. Combined administration of PI3K and MEK inhibitors via fucoidan-based nanoparticles is currently under investigation in pre-clinical models of different cancers.

In summary, our tumour-specific drug delivery strategy shows promising anti-tumour activity in pre-clinical models of HNSCC and could be readily exploited in the clinic to treat patients with this malignancy. In combination with RT, tumour-specific PI3K inhibition may result in enhanced anti-tumour activity without adverse events caused by systemic PI3K inhibition.

## Methods

**Reagents.** Novartis Pharmaceuticals provided BYL719. For *in vitro* assays, all drugs were dissolved in dimethyl sulfoxide. For *in vivo* experiments, BYL719 was dissolved in sterile water, 0.5% carboxymethylcellulose, and 0.2% Tween-80.

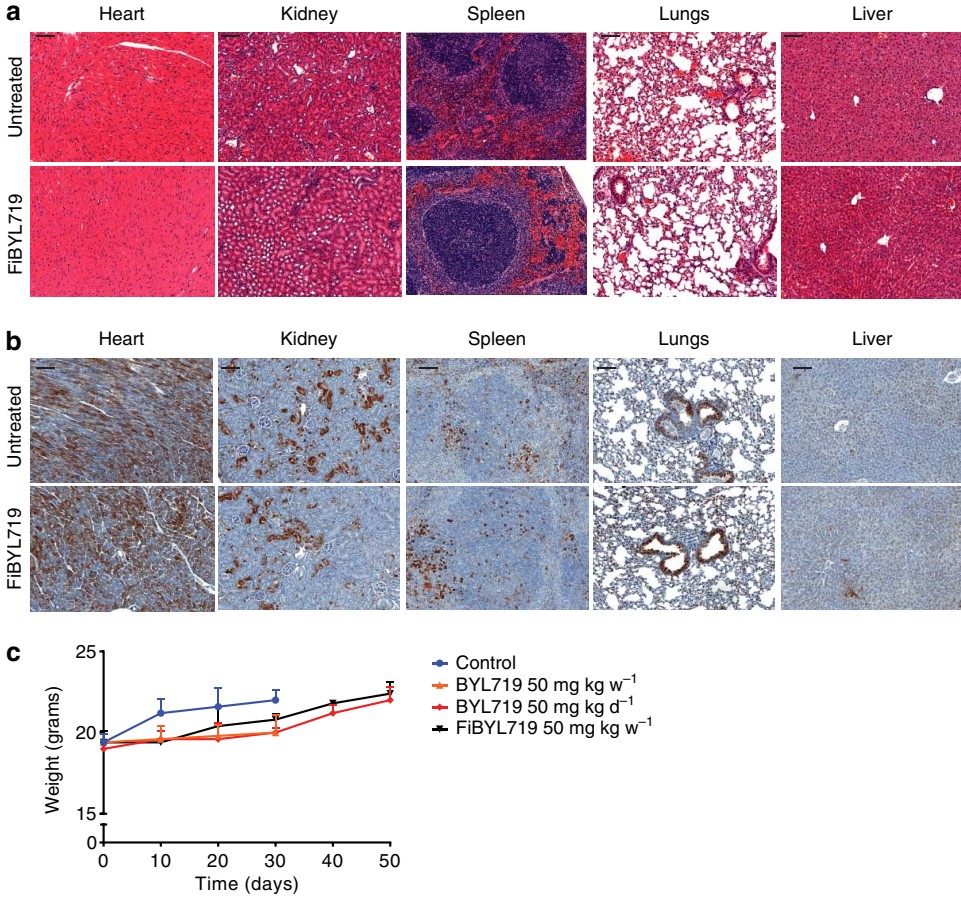

**Figure 5 | Systemic toxicity profiles of FiBYL719 nanoparticles.** (**a**) Representative images of H&E staining in different organs of mice treated with bi-weekly i.v. injections of 25 mg kg$^{-1}$ FiBYL719 for 14 days ($n = 2$), scale bar, 100 μm. (**b**) Representative images of immunohistochemical stains of pS6 in organs of mice treated bi-weekly with i.v. injections of 25 mg kg$^{-1}$ FiBYL719 for 14 days ($n = 2$), scale bar, 100 μm. (**c**) Body weight of tumour-bearing mice treated with oral administration of either 50 or 7 mg kg$^{-1}$ BYL719 daily, or i.v. injections of 25 mg kg$^{-1}$ FiBYL719 bi-weekly ($n = 10$). Error bars indicate mean ± s.d.

**Preparation of FiBYL719 and DexBYL719 nanoparticles.** An aliquot of 0.1 ml of BYL719 dissolved in dimethyl sulfoxide (25 mg ml$^{-1}$) was added drop-wise (20 μl per 15 s) to a 0.6 ml aqueous polysaccharide solution (15 mg ml$^{-1}$) containing IR820 (2.5 mg ml$^{-1}$) and 0.05 mM sodium bicarbonate. An aliquot of 0.1 ml of 8-arm PEG-amine dissolved in water (Creative Peg Works, 20 kD, 5 mg ml$^{-1}$) was added drop-wise to the mixture followed by centrifugation (20,000$g$, 30 min). The nanoparticle pellet was re-suspended in 1 ml of sterile PBS. The suspension was sonicated for 10 s with a probe tip ultrasonicator at 40% intensity (Sonics inc). The nanoparticles were lyophilized in a 5% saline/sucrose solution. Encapsulation efficiency was quantified by ultraviolet–visible spectrophotometer measurement, HPLC-ultraviolet–visible and F-NMR spectroscopy (Supplementary Table 2).

**Nanoparticle characterization.** Dynamic light scattering and zeta potential measurements were conducted in PBS using a Zetasizer Nano ZS (Malvern). Scanning electron microscopy was conducted using a Zeiss Supra 25 Field Emission scanning electron microscope. Samples were prepared by gold sputtering and critical point drying.

**Binding of nanoparticles to P-selectin-expressing endothelial cells.** Bovine aortic endothelial cell monolayers in 24-well plates were pre-incubated with TNFα (50 ng ml$^{-1}$) for 20 min before adding nanoparticles. Control cells were left untreated. The cells were then incubated with 20 μg ml$^{-1}$ of FiBYL719 nanoparticles for 45 min and another 15 min with CellMask Green (Life Technologies) for membrane staining and Hoechst 33,342 (Life technologies) for nuclear staining. The cells were rinsed twice with PBS. Similar method was used for the binding of nanoparticles to irradiated cells. Irradiation of cultured cells was carried out in Shepherd Mark I irradiator containing a $^{137}$Cs source at a rate of 2.08 Gy min$^{-1}$.

Images were acquired with an inverted Olympus IX51 fluorescent microscope, equipped with XM10IR Olympus camera and Excite Xenon lamp. Similar exposure times and excitation intensities were applied throughout all experiments. Filter sets:

cell membrane: $ex$ 488 nm, $em$ 525 nm, nucleus: $ex$ 350 nm, $em$ 460 nm, IR820 dye in particles: $ex$ 780 nm, $em$ 820 nm. Images were processed with ImageJ software.

**Cells and cell culture.** Cal-33 cells were obtained directly from American Type Culture Collection. All cell lines were maintained in humidified incubators at 37 °C in Dulbecco's modified Eagle's medium/Ham's F-12 1:1. Cell culture media was supplemented with 10% heat-inactivated foetal calf serum, 2 mM L-glutamine, penicillin (20 U ml$^{-1}$), and streptomycin (20 μg ml$^{-1}$).

**Determination of *PIK3CA* mutation and copy number status.** *PIK3CA* mutation and amplification status information for Cal-33 cell line was obtained from the Cancer Cell Line Encyclopaedia (www.broadinstitute.org/ccle). Patient-derived xenografts were sequenced using the MSK-IMPACT next-generation sequencing platform[39]. Amplification was defined as greater than or equal to four copies of the *PIK3CA* gene.

**Bio distribution studies of FiBYL719 and DexBYL719 nanoparticles.** Cal-33 and cells were subcutaneously implanted (1 × 10$^6$ cells per injection) in both hind limbs of nude mice. The tumour models were used for bio distribution and tumour growth studies when the tumour size reached 0.5 cm in diameter. Irradiation of the tumours was conducted by giving 4 Gy doses using an X-RAD 225 Cx Micro-Irradiator (Precision X-Ray) with a 50 cm source-to-skin distance. Mice were lightly sedated with ketamine (0.1 mg g$^{-1}$) and xylazine (0.02 mg g$^{-1}$). Only tumour, surrounding skin and subcutaneous tissues were exposed using a specialized lead jig. All of the animal studies were conducted according to protocols approved by the MSKCC Institutional Animal Care and Use Committee.

***In vivo*** **and** ***ex vivo*** **imaging.** Imaging was done 24 h after treatment with 25 mg kg$^{-1}$ of the nanoparticles labelled with IR820 and injected via the tail vein. Mice were anaesthetised with Isoflurane for *in vivo* imaging or killed with

$CO_2$ and then dissected for *ex vivo* organ bio distribution imaging. Images were taken with an IVIS imaging system (Xenogen Corp., Hopkinton, MA). Radiance (photons/ $s^{-1}$ $cm^{-2}$) was calculated for the tumour region of interest using LivingImage V4.2 software.

**In vivo xenograft studies.** All *in vivo* studies were conducted according to the Memorial Sloan Kettering Research Animal Resource Center approved protocols. Six-week-old Nu/Nu mice were ordered from Harlan Laboratories. For cell line-derived xenograft studies, mice were injected bilaterally with $5 \times 10^5$ cells suspended in 200 ul of culture media and Madrigal (BD Biosciences) mixed in a 1:1 ratio. After tumours reached $\sim$100–200 $cm^3$, mice were randomized into treatment arms with 8–10 tumours per group. BYL719 (7 mg $kg^{-1}$ or 50 mg $kg^{-1}$) was dissolved in a vehicle containing 0.5% methylcellulose with 0.2% Tween-80 and was administered via daily oral gavage. FiBYL719 (25 mg $kg^{-1}$ bi-weekly) was given i.v. Tumour volumes were calculated as $(\pi/6) \times$ length $\times$ width$^2$.

For PDX studies, mice were implanted with a tumour obtained from a patient with larynx squamous cell carcinoma. The tumour DNA was sequenced using MSK-IMPACT and found to have a *PIK3CA* H1047R mutation. After tumours reached approximately 100–200 $cm^3$, mice were randomized into treatment arms with 8–10 tumours per group. BYL719 (7 or 50 mg $kg^{-1}$) and FiBYL719 (25 mg $kg^{-1}$) were administered via daily oral gavage or bi-weekly i.v., respectively. Tumours were irradiated with 4 Gy daily on days 1–5 using an X-RAD 320 X-ray system with appropriately sized lead shields. Tumour volumes were calculated as $(\pi/6) \times$ length $\times$ width$^2$.

**Orthotopic HNSCC model.** Cal-33 cells were maintained as a monolayer culture in Dulbecco's Modified Eagle's medium supplemented with 10% foetal bovine serum in a humidified incubator at 37 °C in an atmosphere containing 5% $CO_2$. After the cells have reached confluence, they were transected with a plasmid encoding for both GFP and luciferase for labelling. Cells were then collected and injected into the ventro-lateral aspect of the tongues of nude mice. Approximately $5 \times 10^5$ Cal-33 cells were injected in each mouse and the viability of the injected cells was verified with bioluminescence imaging following intraperitoneal injection of D-Luciferin (50 mg $ml^{-1}$). Tumours were allowed to grow until day 7. For the survival experiment, mice were divided randomly into groups and treated with either oral BYL719 (7 and 50 mg $kg^{-1}$ per day), i.v. FiBYL719 50 mg $kg^{-1}$ per week or vehicle. After treatment, mice were monitored daily for 30 days. Mice were killed using $CO_2$, their tongues were collected, and tumours were examined. The Kaplan–Meier method was used to evaluate survival.

**Serum biochemistry assays.** Whole blood samples were collected into designated tubes from mice at different time points following treatment. Immediately after collection serum was extracted from the samples and stored at − 20 °C until processing. To determine serum fasting glucose levels (mg $dl^{-1}$), we used the hexokinase G-6-PDH method assay (Beckman Coulter). To determine serum insulin levels (μIU/ ml), we used the Access Ultrasensitive Insulin assay (Beckman Coulter).

**Complete blood count.** Whole blood samples were collected into designated tubes from mice 24 h following treatment. Complete blood count was performed using the Hemavet 1700 Haematology System (Drew Scientific Group).

**Immunohistochemistry.** The immunohistochemical detection was performed at the Molecular Cytology Core Facility of Memorial Sloan Kettering Cancer Center using Discovery XT processor (Ventana Medical Systems). All the tissues were collected from mice at different time points following treatment and were fixed in 4% PFA overnight. Fixed tissues were dehydrated and embedded in paraffin before 5 μm sections were put on slides. The tissue sections were deparaffinized with EZPrep buffer (Ventana Medical Systems), antigen retrieval was performed with CC1 buffer (Ventana Medical Systems) and sections were blocked for 30 min with Background Buster solution (Innovex) or 10% normal rabbit serum in PBS (for P-selectin staining). P-selectin sections were incubated with antibodies against P-selectin (R&D Systems, cat# AF737, 5 ugml$^{-1}$) for 5 h, followed by 60 min of incubation with biotinylated rabbit anti-goat IgG (Vector, cat #BA-5000) at 1:200 dilution. pS6 sections were blocked with avidin/biotin block for 12 min, followed by incubation with pS6 antibodies (Cell Signalling, cat# 4370, 1 μg ml$^{-1}$) for 5 h, followed by 60 min incubation with biotinylated goat anti-rabbit IgG (Vector labs, cat#PK6101) at 1:200 dilution. TUNEL sections were incubated with TUNEL antibodies (Vector, cat# VP-K451, 0.4 μg ml$^{-1}$) for 5 h, followed by 60 min incubation with biotinylated goat anti-rabbit IgG (Vector labs, cat#PK6101) at 1:200 dilution. CD-31 sections were incubated with CD-31 antibodies (Dianova, cat# DIA-310, 1 μgml$^{-1}$) for 5 h, followed by 60 min incubation with biotinylated rabbit anti-rat IgG (Vector labs, cat#PK-4004) at 1:200 dilution. Detection was performed with a DAB detection kit (Ventana Medical Systems) according to the manufacturer's instructions, followed by counterstaining with haematoxylin (Ventana Medical Systems) and cover slipped with Permount (Fisher Scientific).

**Immunofluorescence.** The immunofluorescence staining was performed at the Molecular Cytology Core Facility of Memorial Sloan Kettering Cancer Center using Discovery XT processor (Ventana Medical Systems). All the tissues were collected from mice as described in the immunohistochemistry section. Sections were maintained for 10 min on ice and then blocked with 1% BSA in TBS-T for 1 h at room temperature, followed by incubation with γH2AX (1:500, Millipore) antibodies in 1% BSA in TBS-T for 2 h at room temperature. The slides were then washed with TBS-T and incubated with Alexa 555 anti-rabbit and Alexa 488 anti-mouse secondary antibodies (both 1:1,000, Life Technologies) at room temperature. The slides were washed with TBS-T, rinsed in ddH20 and cover slipped with ProLong with DAPI (Life Technologies). Slides were digitally scanned with Pannoramic Flash scanner (3DHistech, Hungary) using 20 × /0.8NA objective. The images were then analysed using Metamorph software (Molecular Devices, PA); briefly, nuclear regions were segmented using the DAPI channel, and the number of foci was counted using spot detection.

**Western blotting.** Specimens were collected at different time points after mice have been killed and immediately flash frozen in liquid nitrogen. The specimens were then lysed in ice-cold radioimmunoprecipitation buffer supplemented with phosphatase inhibitor cocktails (Complete Mini and PhosphoStop, Roche), centrifuged and the supernatant was removed for protein quantification (Pierce BCA Protein Assay Kit, Thermo Scientific). Twenty-five to 50 μg of protein was loaded into NuPAGE 4–12% bis-tris gels (Life Technologies) and resolved via electrophoresis, then transferred to Immobilon transfer membranes (Millipore). Membranes were blocked in 5% BSA in TBS-T for 1 h before overnight incubation in primary antibody at 4 °C, and incubated in either mouse or rabbit horseradish peroxidase-conjugated secondary antibodies (1:50,000, Amersham Biosciences) in 2% BSA in TBS-T for 1 h. Membranes were imaged using SuperSignal West Femto Chemiluminescent Substrate (Thermo Scientific) and images were captured using a GBOX camera system. Antibodies used are listed in Supplementary Table 4. Most significant uncropped scans are show in Supplementary Fig. 5.

**Image analysis.** All automated image analysis was performed using FIJI. Custom macros were created by the Molecular Cytology Core Facility staff. Analysis parameters and thresholds remained consistent throughout experimentation.

**Statistical analysis.** Comparisons between groups were evaluated using the nonparametric Mann–Whitney *U*-test or the one-way ANOVA with *post hoc* Tukey test, and $P < 0.05$ was considered significant. Survival curves were estimated using the Kaplan–Meier method and compared by the log-rank test. All statistical analyses were performed using the SPSS software version 21.0 (SPSS Inc., Chicago, IL) or Prism version 6 software (GraphPad Software, Inc., La Jolla, CA).

**Data availability.** Detailed genomic information about the cell lines can be obtained from the Cancer Cell Line Encyclopaedia (www.broadinstitute.org/ccle). All other relevant data are available from the authors.

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

## Acknowledgements

This work was supported by the Cancer Center Support Grant (P30 CA008748), the NIH Director's New Innovator Award (DP2-HD075698), New York State Department of Health DOH01-C30315GG-3450000, the Center for Experimental Therapeutics of Memorial Sloan Kettering Cancer Center, Mr William H. and Mrs Alice Goodwin and the Commonwealth Foundation for Cancer Research, the Anna Fuller Fund, the Louis V. Gerstner Jr Young Investigator's Fund, the Frank A. Howard Scholars Programme, the Imaging and Radiation Sciences Programme and the Center for Metastasis Research at Memorial Sloan Kettering Cancer Center. Y.S. was supported by the Center for Metastasis Research Scholars Fellowship Programme. We thank the following core facilities at MSKCC: Molecular Cytology, Electron Microscopy and Small Animal Imaging. We thank Sho Fujisawa for assistance with digital imaging and data analysis and Afsar Barlas for immunofluorescence staining.

## Author contributions

A.M., Y.S. and M.S. designed the research; A.M., Y.S., J.S., S.B., V.K.R. and J.S. performed experiments; A.M., S.B. and M.S. collected samples and obtained clinical data; A.M., Y.S., A.H.-F., J.L.H., D.A.H. and M.S. analysed data; S.N.P., J.H.H., J.B., M.S., A.H.-F. and D.A.H. supervised the research. Y.S., J.H.H., D.A.H., A.H.-F. and M.S. wrote the paper.

## Additional information

**Competing financial interests**: J.B. has consulted for Novartis Pharmaceuticals and serves on the board of Infinity Pharmaceuticals.

**Reprints and permission** information is available online at yhttp://npg.nature.com/reprintsandpermissions/

**How to cite this article**: Mizrachi, A. *et al.* Tumour-specific PI3K inhibition via nanoparticle-targeted delivery in head and neck squamous cell carcinoma. *Nat. Commun.* **8**, 14292 doi: 10.1038/ncomms14292 (2017).

