## [Peer Review File · Nature Communications]

Reviewers' comments:

Reviewer #1 (Remarks to the Author):

In the manuscript entitled "Tumor-specific PI3K inhibition via nanoparticle targeted delivery in head and neck squamous cell carcinoma", Aviram and co-authors prepared a P-selectin targeted nanoparticle (FiBYL719) for tumor targeted BYL719 delivery. They showed that the FiBYL719 resulted in tumor selective accumulation and effective anti-tumor activity, also synergistic effect with radiation therapy. Importantly, the acute and chronic metabolic side effects seen after BYL79 treatment are abrogated when using the FiBYL719 treatment. The work is well designed, and the manuscript is well-organized. The following are several points need to be considered:

1. It's improper to use "synthesize" for the nanoparticle preparation. The drugs are encapsulated inside fucoidan physically, there is no chemical reaction, so it's not "synthesized".
2. A scheme to show the preparation process of the FiBYL719 and DexBYL719 will be helpful.
3. What's the drug loading content and efficiency of FiBYL719 and DexBYL719? Also, please provide an in vitro drug release result.
4. It's impressive that the targeted FiBYL719 showed obvious selectivity to tumor, compared with DexBYL719 and other organs (Figure 1a,b). But I think a more detailed in vivo data is needed to confirm the targeting mechanism of the FiBYL719 to P-selection on the vasculature or tumor cells. I recommend an immunofluorescence analysis to show the exact distribution of FiBYL719 inside the tumor tissue.
5. I'm concerned with the systemic toxicity of the anti-tumor therapy test shown in Figure 2e, f, g. Is there any body weight lost? Is there damage to normal organs after encapsulating the BYL719 inside nanoparticles? Although the author measured the impact on hyperglycemia in healthy mice, I still wonder the possible side effects to normal organs during treatment.
6. Please give a careful examination on the manuscript. For example, in Page 10, line 228, "IR820" should be "IR783". In Figure 4, (d) and (e) are reversed.

Reviewer #2 (Remarks to the Author):

HNSCC show increased PIK3CA mutations, but therapy with PI3K inhibitors is limited by side effects including hyperglycemia. The investigators used p-selectin targeted nanoparticles to enhance tumor-specific delivery of PI3Ka isoform specific drug to HNSCC PDX and orthotopic xenograft models. Compared with free drug, they show convincingly that the nanoparticle vehicle enhances inhibition of tumor growth, molecular biomarkers, and enhances cell cytotoxic effects and survival alone and in combination with radiation, previously observed with PI3K-mTOR inhibitors. The hyperglycemia and insulinemia seen with free drug was attenuated over a prolonged administration. This may represent a significant advantage in further studies in these tumors.

Some minor concerns:

The p53 mutation and p-AKT status of the models should be reported along with the PIK3CA status, as these have been reported to affect drug sensitivity in HNSCC (Herzog A Clin Cancer Res 2013; Mohan S Clin Cancer Res 2015).

These variables along with p-ERK staining reported may help explain the incomplete effects seen, please address this and whether targeted delivery of MEK inhibitors could also be enhanced in the discussion.

Suppl Fig 3e is not of sufficient size or resolution to convincingly see the P-selectin staining claimed in the orthotopic tongue tumors.

Reviewer #1:

In the manuscript entitled “Tumor-specific PI3K inhibition via nanoparticle targeted delivery in head and neck squamous cell carcinoma”, Aviram and co-authors prepared a P-selectin targeted nanoparticle (FiBYL719) for tumor targeted BYL719 delivery. They showed that the FiBYL719 resulted in tumor selective accumulation and effective anti-tumor activity, also synergistic effect with radiation therapy. Importantly, the acute and chronic metabolic side effects seen after BYL79 treatment are abrogated when using the FiBYL719 treatment. The work is well designed, and the manuscript is well organized.

We thank the Reviewer for considering our work well designed and the manuscript well organized.

The following are several points need to be considered:

1. It's improper to use “synthesize” for the nanoparticle preparation. The drugs are encapsulated inside fucoidan physically, there is no chemical reaction, so it's not “synthesized”.

Following the Reviewer's observation, we have corrected this throughout the manuscript.

2. A scheme to show the preparation process of the FiBYL719 and DexBYL719 will be helpful.

We agree with the Reviewer and we now show such a scheme in Supplementary Figure 2 of the new version of the manuscript (also shown below).

3. What's the drug loading content and efficiency of FiBYL719 and DexBYL719? Also, please provide an *in vitro* drug release result.

The drug loading content and efficiency of FiBYL719 and DexBYL719 has now been added in Supplementary Table 2 (see also below).

Nanoparticle	Drug content	Loading efficiency
FiBYL719	22%	82%±5%
DexBYL719	20%	79%±4%

In addition, an *in vitro* drug release profile is now shown in panel e of Supplementary Figure 3 (see also below).

4. It's impressive that the targeted FiBYL719 showed obvious selectivity to tumor, compared DexBYL719 and other organs (Figure 1a,b). But I a more detailed in vivo data is needed to confirm targeting mechanism of the FiBYL719 to P-selection on the vasculature or tumor cells. I recommend an immunofluorescence analysis to show the exact distribution of FiBYL719 inside the tumor tissue.

Drug release profile of BYL719 from FiBYL719 nanoparticles over time at PBS buffers of pH 5.5 and 7.4

with think the

Following the Reviewer's advice, we have analyzed by immunofluorescence the distribution/localization of the labeled-FiBYL719 nanoparticles within the H22 patient-derived xenografts. Given the rapid photo bleaching of the labeling dye, we had to optimize a

new protocol based on confocal imaging on frozen tissue section. Consistent with our previous results, we found that the labeled FiBYL719 nanoparticles distributed throughout the

Representative immunofluorescence staining for P-selectin (green), NIR (red) and DAPI (blue) in H22 xenografts 24h after treatment with either FiBYL719 or 4 Gy followed by FiBYL719, scale bars, 50 μm.

tumor tissue and more so after administration of radiation. This is now shown in panel g of Figure 1 (also shown above).

5. I'm concerned with the systemic toxicity of the anti-tumor therapy test shown in Figure 2e, f, g. Is there any body weight lost? Is there damage to normal organs after encapsulating the BYL719 inside nanoparticles? Although the author measured the impact on hyperglycemia in healthy mice, I still wonder the possible side effects to normal organs during treatment.

The Reviewer's concern is understandable. Although no weight loss was observed following treatment with FiBYL719 (Figure 5 panel c, also shown below), we have conducted new *in vivo* studies

to specifically rule out possible damage to normal organs of animals treated for 14 days with the drug-containing nanoparticles. As shown in Figure 5 panel a (also shown on the left), no morphologic changes suggestive of tissue damage were observed in the heart, kidney, spleen, lungs or liver of the examined necropsy specimens. Moreover, no inhibition of pS6 was seen in any of these organs.

damage were observed in the heart, kidney, spleen, lungs or liver of the examined necropsy specimens. Moreover, no inhibition of pS6 was seen in any of these organs.

6. Please give a careful examination on the manuscript. For example, in Page 10, line 228, “IR820” should be “IR783”. In Figure 4, (d) and (e) are reversed.

We thank the reviewer for pointing this out and have re-examined the manuscript to address such typos.

Reviewer #2:

HNSCC show increased PIK3CA mutations, but therapy with PI3K inhibitors is limited by side effects including hyperglycemia. The investigators used p-selectin targeted nanoparticles to enhance tumor-specific delivery of PI3Ka isoform specific drug to HNSCC PDX and orthotopic xenograft models. Compared with free drug, they show convincingly that the

Systemic toxicity profile of FiBYL719 nanoparticles. (a) Representative images of H&E staining in different organs of mice treated with bi-weekly i.v. injection of 25 mg/kg FiBYL719 for 14 days ($n=2$), scale bar, 100 μm . (b) Representative images of immunohistochemistry staining for pS6 in different organs of mice treated with bi-weekly i.v. injection of 25 mg/kg FiBYL719 for 14 days ($n=2$), scale bar, 100 μm . (c) Body weight (grams) of tumor-bearing mice treated with daily oral administration of either 50 mg/kg or 7 mg/kg BYL179, or bi-weekly i.v. injection of 25 mg/kg FiBYL719 ($n = 10$).

nanoparticle vehicle enhances inhibition of tumor growth, molecular biomarkers, and enhances cell cytotoxic effects and survival alone and in combination with radiation,

previously observed with PI3K-mTOR inhibitors. The hyperglycemia and insulinemia seen with free drug was attenuated over a prolonged administration. This may represent a significant advantage in further studies in these tumors.

We thank the Reviewer for considering our results convincing and a potential significant advancement in the field.

Some minor concerns:

The p53 mutation and p-AKT status of the models should be reported along with the PIK3CA status, as these have been reported to affect drug sensitivity in HNSCC (Herzog A Clin Cancer Res 2013; Mohan S Clin Cancer Res 2015).

We now report both p53 status and basal levels of pAKT in the tumor models listed in our manuscript. These data are shown in Supplementary Table 1 (also shown below). We also added representative images of IHC staining for pAKT in different HNSCC models (Supplementary Fig. 1a)

PDX	Source	p16	HPV	PIK3CA	TP53	pAKT	P-selectin
H16	Tonsil	+	-	E542K	WT	+	-
H22	Larynx	-	-	H1047R	R248Q	+	+
H30	Tonsil	+	+	E542K	WT	-	+
H31	UKP	+	+	WT	WT	+	+
H33	BOT	+	-	V344G	WT	+	+

These variables along with p-ERK staining reported may help explain the incomplete effects seen, please address this and whether targeted delivery of MEK inhibitors could also be enhanced in the discussion.

We agree with the Reviewer that feedback activation of ERK can be a plausible explanation for the incomplete effects of PI3K inhibition *in vivo*. As a matter of fact, several reports showed that concomitant inhibition of MEK and PI3K/mTOR results in superior antitumor activity in head and neck cells/tumors and in other tumor models (Mohan et al. Clin Cancer Res 2015, Serra et al. Oncogene, 2011, Garcia-Garcia et al. Clin Cancer Res 2015).

We have now addressed this topic in page 6 of the revised manuscript adding the following statements: “It remains to be elucidated whether concomitant inhibition of ERK by either systemic or nanoparticle-based treatment with MEK inhibitors would further enhance the antitumor activity of BYL719. Previous reports testing the efficacy of PI3K/mTOR blockade in combination with MEK inhibition in HNSCC (Mohan et al. CCR) and other tumor (Serra et al. Oncogene, Garcia-Garcia et al. CCR) models suggest that this may be the case”.

Suppl Fig 3e is not of sufficient size or resolution to convincingly see the P-selectin staining claimed in the orthotopic tongue tumors.

We have now corrected this.

REVIEWERS' COMMENTS:

Reviewer #1 (Remarks to the Author):

The author has given point by point responses to the previous review, and supplied required data. They have satisfied our concerns. Therefore, I think the paper can be accepted in the current version.

Reviewer #2 (Remarks to the Author):

From my perspective, the authors have fully addressed the reviewer's comments.